# OpenReview forum: "Working Memory Attack on LLMs"
_ICLR.cc/2025/Workshop/BuildingTrust — BuildingTrust_

### Official Review · Reviewer_riWm · 2025-02-25
**A novel working memory attack that bypasses safety mechanisms on SOTA LLMs**

**Rating:** 7
**Confidence:** 4

**Review:**

This paper explores the vulnerabilities of LLMs within the scope of in-context learning. The authors develop a novel multi-task methodology inspired from neuroscience to systematically measure and quantify working memory overload in LLMs. They present a new attack vector that exploits working memory overload to bypass safety mechanisms in state-of-the-art LLMs, achieving high attack success rates across multiple models. They also demonstrate the transferability of these attacks, showing that higher-capability LLMs can be used to craft working memory overload attacks targeting other models.

**Clarity:** The paper is generally well-written and easy to follow. The authors provide clear explanations of their methodology and findings. However, some sections could benefit from more detailed explanations, especially those describing the parallels to neuroscience as this is not a field most readers in this field may be familiar with.

**Strengths:**
- The paper introduces a novel and interesting approach to studying the vulnerabilities of LLMs.
- The use of a multi-task methodology inspired by neuroscience is a creative and effective way to measure the working memory limitations of LLMs.
- The paper provides empirical evidence that working memory overload can be used to bypass safety mechanisms in LLMs.
- The authors demonstrate the transferability of their attacks, which highlights the potential for these attacks to be used in the wild.

**Weaknesses:**
- The paper primarily focuses on bypassing specific safety mechanisms. A broader exploration of various safety protocols would enhance the paper's impact.
- The methodology's reliance on specific task types might limit its generalizability. Further validation with diverse tasks would be beneficial.
- While the attack is shown to be effective, a more in-depth explanation of the underlying mechanism would improve understanding and reproducibility

**Areas of Improvement:**
- Discuss potential defenses and mitigations against the proposed attack.
- Include a more thorough analysis of the computational resources required for the attack.
- Provide more detailed explanations of the neuroscience concepts that they draw upon.
- Discuss the potential implications of their findings for the development of more robust and secure LLMs.

Overall, the paper introduces a novel new attack that is effective on state-of-the-art LLMs by attacking the working memory space and bypassing safety mechanisms. The authors should address the weaknesses and limitation of this attack along with having further discussions on the implication and potential next steps.

---

### Official Review · Reviewer_1Xmy · 2025-03-01
**Review: Working Memory Attack on LLMs**

**Rating:** 7
**Confidence:** 3

**Review:**

The paper "Working Memory Attack on LLMs" applies ideas about working memory from neuroscience to expose a vulnerability in LLM reasoning. The authors show that progressively increasing task-irrelevant token generation before a final focus "observation task" significantly degrades model performance, providing a series of prompts, with sufficient justification for their choice. This attack vector is said to exploit working memory overload to bypass safety mechanisms in state-of-the-art LLMs. The authors provide a great visual example of this, promting the LLMs to generate visualization generation code in python using tikz, which is a nice touch.

The quality of this research is high, with a well-structured methodology. The authors do focus on the same sequence of memory overload attacks, and it would be nice to see experiments on other types of working memory overloads too. The authors provide sufficient justification for their choices in the appendix which is great. The authors provide a thorough statistical analysis to support their findings, including paired t-tests.

The clarity of the text is acceptable, and the highlight was the visual example. There seemed to be redundant repeated phrasing but this is a non issue, but might be a consideration for the camera ready version. A bigger issue is potential logic issues stating that degraded reasoning implied working memory limitations in LLMs. Working memory had not been well defined enough to claim that the sequence of prompts indeed affected the working memory of the LLM. It would be nice to see more rigour here. Additionally, it would be nice if the authors attempted to find the minimal working memory attack to degrade performance, but that can be left to future work.

The research is highly original, considering the  concept of working memory overload from neuroscience as an attack vector. The authors not only identify a new attack vector but also provide empirical evidence that this vulnerability is widespread across different SOTA LLM architectures. The potential for these attacks to transfer between models is particularly concerning and underscores the need for new safety mechanisms.

The paper lacks a detailed discussion on potential countermeasures against working memory overload attacks and the use of judge LLMs to assess the safety of outputs might introduce biases. The latter is a concern but previous results support the general results anyways. The authors do not discuss the practicality of these attacks: the method of progressively increasing complexity in prompts may not be practical in all scenarios. It would be nice to see a minimal sufficient working memory overload attack in this case.

---

### Official Review · Reviewer_CRtq · 2025-03-02
**Insightful Study of Cognitive Limitations in Language Models**

**Rating:** 8
**Confidence:** 4

**Review:**

**Summary of claims for contribution**

The paper contributes with a new attack vector that exploits a vulnerability in LLMs which the authors call “working memory overload”. This way of attacking LLMs degrades model performance and bypasses safety mechanisms. They demonstrate how this attack can be effective across models and develop an automated attack algorithm to evaluate the safety alignment of different LLMs.

**List strong and weak points of the paper**

The strong points of the paper is that they demonstrate a successful attack that significantly reduces the capability or safety alignment of state of the art LLMs. The evaluations of the attacks are fairly comprehensive, spanning multiple task types and models, thus showing that it is transferrable. They use a well structured experimental methodology to measure and quantify memory overload.  Lastly, connecting this approach to cognitive science provides opportunities for new research directions.

Some of the weaker points include:
* Limited theoretical explanation for why this vulnerability exists in transformer architectures
* Lack of justification for why this is analogous to human working memory, the reason LLMs underperform on these tasks can be due to other factors
* Minimal exploration of potential defences against these attacks
* There could have been more exploration on which specific aspects of the attack are most critical to its success and which models are most vulnerable and why (more ablation studies)

Recommendation: Accept

The paper introduces a novel attack vector for LLMs that has very high success rate and exposes a vulnerability that needs further study. It produces strong empirical evidence and covers multiple LLMs, showing that it transfers across models. These types of attacks are of great relevance for AI safety and need further attention.

---

### Decision · Program_Chairs · 2025-03-04

**Decision:**

Accept

**Comment:**

This paper examines LLM vulnerabilities in in-context learning, introducing a neuroscience-inspired multi-task method to quantify working memory overload. It reveals a novel attack exploiting this overload to bypass safety mechanisms, achieving high success rates and transferability across models.